# Correlation between Chemical Composition and Antifungal Activity of *Clausena lansium* Essential Oil against *Candida* spp.

**DOI:** 10.3390/molecules24071394

**Published:** 2019-04-09

**Authors:** Xiaowen He, Lantong Zhang, Jinping Chen, Jinlei Sui, Guohui Yi, Jinyan Wu, Yinzheng Ma

**Affiliations:** 1Public Research Laboratory, Hainan Medical University, Haikou 571199, Hainan, China; xiaowen_he@126.com (X.H.); 15595767678@163.com (J.C.); sui3com@126.com (J.S.); guohuiyi6@163.com (G.Y.); jinyanzi0720@126.com (J.W.); 2School of Pharmacy, Hebei Medical University, Shijiazhuang 050017, Hebei, China; zhanglantong@263.net; 3School of Public Health, Hainan Medical University, Haikou 571199, Hainan, China

**Keywords:** correlation, *Clausena lansium*, essential oil, antifungal activity, *Candida* spp.

## Abstract

Essential oils (EOs) have been shown to have a diversity of beneficial human health effects. *Clausena* is a large and highly diverse genus of plants with medicinal and cosmetic significance. The aim of this study was to analyze the composition of *Clausena lansium* EOs and to investigate their potential antifungal effects. The chemical compositions of *Clausena lansium* EOs obtained by hydrodistillation were analyzed by gas chromatography-mass spectrometry (GC-MS). A total of 101 compounds were identified among the diverse extracts of *C. lansium*. EOs of leaves and pericarps from different cultivars (Hainan local wampee and chicken heart wampee) collected in Hainan (China) were classified into four clusters based on their compositions. These clusters showed different antifungal activities against five *Candida* species (*C. albicans*, *C. tropicalis*, *C. glabrata*, *C. krusei* and *C. parapsilosis*) using the disc diffusion method. *Clausena lansium* EOs of pericarps displayed noteworthy antifungal activitives against all the tested *Candida* strains with inhibition zone diameters in the range of 11.1–23.1 mm. EOs of leaves showed relatively low antifungal activities with inhibition zone diameters in the range of 6.5–22.2 mm. The rank order of antifungal activities among the four EO clusters was as follows: Cluster IV> Cluster III > Cluster I ≥ Cluster II. These results represent the first report about the correlation between chemical composition of *C. lansium* EOs and antifungal activity. Higher contents of β-phellandrene, β-sesquiphellandrene and β-bisabolene in EOs of pericarps were likely responsible for the high antifungal activity of Cluster IV EOs. Taken together, our results demonstrate the chemical diversity of *Clausena lansium* EOs and their potential as novel antifungal agents for candidiasis caused by *Candida* spp. Furthermore, the obtained results showing a wide spectrum of antifungal activities provide scientific evidence for the traditional use of these plants.

## 1. Introduction

Natural plant products, especially essential oils (EOs), have many beneficial biological effects, such as antibacterial, anti-inflammatory, antitumor and analgesic activities. EOs are volatile aromatic substances found in many plants. The EOs may exist in fruits, seeds, flowers and leaves. They have already significant attention because of their abundance, broad spectrum activities, and diverse mechanisms of action. They are among the most popular natural antimicrobial agents and they have recently gained a great popularity and scientific interest [1,2,3,4].

*Clausena lansium* (Lour.) Skeels, commonly known as wampee, is a tropical species that belongs to the family Rutaceae. It is reported as a native of Southern China, mainly distributed in Hainan, Fujian, Guangxi, Guangdong, Taiwan, etc. [5]. *Clausena lansium* has attracted great attention owing to its extensive pharmacological benefits, including anticancer, antioxidant, antidiabetic, antinociceptive, hepatoprotective, acerebroprotective and anti-trichomonal effects [6,7,8,9]. EOs of *C. lansium* play an important role in the abovementioned benefits [5]. In traditional Chinese medicine, the leaf and pericarp of *C. lansium* are often used to treat different diseases [10]. For example, the leaves of *C. lansium* have been used for treating fever, cough, asthma, malaria, dermatological and rheumatism, etc. When fruit turns from pale green to pale yellow or brownish-yellow around July, it can be eaten along with pericarp. Mature fruits have been used for treating digestive disorders, gastro-intestinal diseases and bronchitis. A likely mechanism is that EOs from *C. lansium* fruits can help release gastrointestinal gas, eliminates stagnation, dissipates heat and relieves pain.

Hainan Province, a tropical region of China, is one of the main geographic regions for growing *Clausena lansium*. *Clausena lansium* with globose fruits is named Hainan local wampee (HLW): the berry fruit is approximately 1.5–2.5 cm in diameter. *Clausena lansium* with broadly ovoid fruits is named chicken heart wampee (CHW), with berries approximately 1.5–2.5 × 2.5–3.5 cm in diameter [11]. The compositions and contents of *C. lansium* EOs from China, Thailand, and Cuba have been studied by hydrodistillation followed by gas chromatography-mass spectrometry (GC-MS) [12,13,14]. However, limited effort has been made to analyze and compare the differences of EOs of HLW and CHW. In addition, due to the increasing number of fungal diseases caused by *Candida* species and insufficient effectiveness of traditionally applied azole agents, new therapeutic strategies are necessary and natural products especially EOs can play an important role in the treatment of infections [15,16]. Therefore, the search of new and effective natural antifungal agents has dramatically increased [17]. The aim of the current study was to comparatively analyze the chemical compositions of leaves and pericarps EOs of *Clausena lansium* of HLW and CHW and to investigate the in vitro antifungal activities against *Candida* species, including *C. albicans*, *C. glabrata*, *C. krusei*, *C. parapsilosis* and *C. tropicalis*. The results of this study can provide experimental basis for further exploitation of *C. lansium*.

## 2. Results and Discussion

### 2.1. Extraction Results and Chemical Composition of EOs

*Clausena lansium* EOs extracted by hydrodistillation were light yellow to yellow in color. The yields ranged from 0.23 to 0.51% (*v*/*w*). The source, cultivar, part, color and yield of EO details are listed in Table 1. There were differences in the color and yield of the eight EOs, which may be related to the geographic source, cultivar, and part of plants, but it is uncertain which is the most important of these factors. It is possible to come to a conclusion from the specific differences of the components. To compare the compositional differences, each EO extract was injected into GC-MS for analysis. Figure 1 shows GC-MS chromatograms of *C. lansium* EOs. The chemical compositions and contents results are shown in Table 2.

According to the GC-MS analysis, components and contents were different in leaves and pericarps of HLW and CHW. A total of 69, 63, 65, 63, 70, 71, 69 and 66 compounds were identified in MCL, MCP, RCL, RCP, MHL, MHP, RHL and RHP, amounting to 98.87%, 96.90%, 99.19%, 98.18%, 98.10%, 98.60%, 96.87% and 96.39% of the total components, respectively. These components were monoterpenes, sesquiterpenes, alcohols, esters, etc. Some samples such as MCL and RCL were dominated by sesquiterpenes, with β-caryophyllene and β-sesquiphellandrene as the main components. MHL and RHL were rich in sesquiterpenes with *cis*-α-santalol and santalol being the most abundant species. There were a large number of monoterpenes in some samples, such as MCP, RCP, MHP and RHP. In all the combined samples, we detected 101 chemicals in the EO extracts. However, the distributions of the 101 chemicals differed widely among the extracts. To investigate the similarity and differences among these extracts in EO profiles, a cluster analysis was conducted using SPSS (version 17.0). The clustering result is shown in the Figure 2.

Samples were clustered into two clusters for the first time, leaves and pericarps. This result indicates that most of the differences were caused by plant parts. The next contributor was plant cultivar, followed by geographic location of where the plants were grown. The difference between Meixiao Village and Rulin Village was the smallest, which is probably because the two places are geographically close (both in Haikou city). Table 3 shows the variation of some important volatile components (%) of *C. lansium* in four clusters.

The domonant EOs in each of the four clusters are listed below:

Cluster I: β-caryophyllene and β-sesquiphellandrene. MCL and RCL belonged to this cluster. These samples were leaves of CHW. The components were rich in β-caryophyllene (21.13%, 23.48%), β-sesquiphellandrene (18.52%, 21.80%), and accompanied by α-bergamotene (9.49%, 10.63%).

Cluster II: *cis*-α-santalol and β-santalol. There were MHL and RHL in this cluster. These samples were leaves of HLW. *cis*-α-Santalol (14.72%, 21.31%), β-santalol (17.31%, 12.24%) and caryophyllene (12.72%, 5.46%) were the most predominant components.

Cluster III: β-phellandrene and β-bisabolene. MHP and RHP were classified into this cluster. They were pericarps of HLW. They contained more β-phellandrene, a relative percentage of 26.13% and 23.15%, respectively. Percentages of β-bisabolene and *cis*-α-santalol were also relatively large (6.59–10.21%).

Cluster IV: β-phellandrene and β-sesquiphellandrene. Pericarps of CHW such as MCP and RCP in this cluster. β-Phellandrene in this cluster had a percentage higher than 30%. The percentages of β-phellandrene were 32.43% and 45.15% in MCP and RCP, respectively. β-Sesquiphellandrene was also relative high, with percentages of 10.89% and 7.35%, respectively.

In previous studies, EOs of *C. lansium* from China [12], Thailand [13], and Cuba [14] were obtained by hydrodistillation and analyzed by GC-MS. Thirty-two components were identified in leaves samples from China. The main components identified in the EO of leaves were β-santalol (35.20%) and bisabolol (13.70%). In both fresh and dried fruit samples from Thailand, fifty-three components were identified, and the main components were sabinene (33.68–66.73%), α-pinene (9.57–13.35%) and 1-phellandrene (5.77–10.76%). For leaf samples from Cuba, seventy compounds were identified. The most prominent components were caryophyllene oxide (16.80%) and (*Z*)-α-santalol (11.70%). The components and percentages were different from different areas, and the differences were much greater in different parts of *C. lansium*. There were some common components in EOs of leaves in the present study and the above reports, such as *cis*-α-santalol, β-bisabolene, and caryophyllene oxide. In contrast, the components of pericarps in our study were very different from those of the fruit samples in previous studies. The reason may be that fruit contains not only pericarp, but also pulp and seed.

In addition to the hydrodistillation method, supercritical fluid extraction (SFE) was used in the extraction of *C. lansium* leaves from Guangdong, China [18]. Thirty-six components were identified and the main components were 4-terpineol (26.94%) and γ-terpinene (14.39%). Chokeprasert et al. [19] analyzed the volatile components of fresh leaves and pericarps in Thailand by headspce (HS) GC-MS. Thirty-nine components were identified in the leaves, and the major components were sabinene (15%) and β-bisabolene (9.88%); thirty components were identified in the pericarps, and the major components were sabinene (69.07%) and α-phellandrene (10.63%). The components and percentages obtained by SFE and HS were quite different from that by hydrodistillation.

In the present study, the differences of the compositions of *Clausena lansium* leaves and pericarps from different cultivars were analyzed and compared for the first time. Cluster analysis result showed that the differences in the components were most significantly contributed by plant parts, followed by cultivar, and with geographic origin contributing the least to the variation.

### 2.2. Antifungal Activity

The antifungal activity of *C. lansium* EOs was examined using the filter paper disc diffusion method. As shown in Table 4, the EOs inhibited the growth of all seven yeast strains tested in our study. The inhibition zone diameters were in the range of 6.5–23.1 mm. The EOs of *Clausena lansium* pericarps showed significant activity against all the *Candida* strains, whereas, EOs of leaves showed relatively poor activity. The RCP EO exhibited the greatest antifungal effect against *C. glabrata*, with an inhibition zone diameter of 23.1 mm. Notably, *C. albicans* 27, a yeast strain which is resistant to FLZ and AMB, can be inhibited by *Clausena lansium* EOs with an inhibition zone diameter of 11.3–15.3 mm. This observation suggests that *Clausena lansium* EOs is active towards certain FLZ and AMB resistant strains.

It can be seen from Table 4, the antifungal activitives efficacy order was as follows: Cluster IV ≥ Cluster III > Cluster I ≥ Cluster II. The two most active EOs against *C. albicans*, *C. glabrata*, *C. krusei* and *C. parapsilosis*, which were pericarps of CHW (Cluster IV) and were all rich in β-phellandrene (32.43%, 45.15%) and β-sesquiphellandrene (10.89%, 7.35%). Inhibition zone diameters were in the range of 12.0–23.1 mm. EOs of pericarps of HLW (Cluster III) were the most active against *C. tropicalis* and *C. albicans* 27 (a clinical strains resistant to FLZ and AMB), with the most predominant components being β-phellandrene (26.13%, 23.15%) and β-bisabolene (7.74%, 10.21%). Inhibition zone diameters were in the range of 11.1–22.1 mm. The percentages of the monoterpene β-phellandrene in the two clusters were all high. There were some differences of compositions between Cluster I and II, but they were all dominated by sesquiterpenes as the main components. The antifungal activities against *C. glabrata* were almost equal to Cluster IVand III with inhibition zone diameters 22.0–22.2 mm. However, the antifungal activities against other six *Candida* strains were relatively low, with inhibition zone diameters 6.5–13.0 mm. Antifungal effect of EOs of CHW was higher than that of HLW in both leaves and pericarps. Comprehensive analysis illustrates the importance of monoterpenes to antifungal activities and there were differences among different types of *Candida* strains. The results agree with those of Białon et al. [20] who suggested that the antifungal potential of lemon essential oils against *Candida* yeast strains was related to the high content of monoterpenoids and the type of *Candida* strains.

There are few studies reporting the antifungal activity of *Clausena lansium* extracts against infectious yeasts. XU et al. [21] reported that the extracts of pericarps of *Clausena lansium* using 95% alcohols had the significant inhibitory effect on *C. albicans* with an inhibition zone diameter of 16.8 mm by the filter paper disc diffusion method. However, the components of this extracts are not yet clear. In the present study, we obtained *Clausena lansium* EOs by hydrodistillation and determined the chemical compositions by GC-MS. In all the tested EOs of pericarps, β-phellandrene was determined to be present at the highest percentage which could be mainly responsible for the antifungal effects. β-Sesquiphellandrene and β-bisabolene were also important compositions which were possible to have antifungal activities. Perigo et al. [22] studied the chemical compositions and antibacterial activities of *Piper* species from distinct rainforest areas in Southeastern Brazil, and found that higher contents of β-phellandrene were positively correlated to wide spectrum antibacterial activity. Antibacterial activity of essential oils of *Tripleurospermum disciforme* was reported that the reason for higher antibacterial effect is the presence of β-farnesene and β-sesquiphellandrene [23]. The essential oils of *Bocageopsis pleiosperma Maas* rich in β-bisabolene were reported to have high antifungal activities [24]. Their results are consistent with ours. Taken together, we speculate that antifungal activities in our study may be attributed to *C. lansium* EOs of pericarps rich in β-phellandrene, β-sesquiphellandrene and β-bisabolene.

However, antifungal activity may not entirely depend on the main chemical components. Other chemical components in lower concentrations may also have antifungal activity or have synergistic effects with other components. Therefore, it is necessary to study antifungal activity of the component alone or in combination with others, in order to further elucidate the material basis of antifungal activity of *C. lansium* EOs against *Candida* spp. It may provide a basis for the development of *C. lansium* EOs as new antifungal agents with high efficiency, broad spectrum, low toxicity and low cost.

## 3. Conclusions

This study described the chemical compositions of the EOs from leaves and pericarps of *Clausena lansium* (Lour.) Skeels, where a total 101 compounds were identified. EOs of leaves and pericarps of HLW and CHW in Hainan, China were classified into four clusters by cluster analysis. The major components of cluster I were caryophyllene (21.13%, 23.48%), β-sesquiphellandrene (18.52%, 21.80%) and α-bergamotene (9.49%, 10.63%); Cluster II: *cis*-α-santalol (14.72%, 21.31%), santalol (17.31%, 12.24%) and caryophyllene (12.72%, 5.46%); Cluster III: β-phellandrene (26.13%, 23.15%) and β-bisabolene (7.74%, 10.21%); Cluster IV: β-phellandrene (32.43%, 45.15%) and β-sesquiphellandrene (10.89%, 7.35%). Correlation between chemical composition and antifungal activity is an important finding. EOs of *Clausena lansium* pericarps showed higher antifungal activity than that of leaves against all the tested *Candida* strains, and EOs of CHW had better inhibiting effect than HLW. Higher contents of β-phellandrene, β-sesquiphellandrene and β-bisabolene may be positively correlated to antifungal activity. The present results increase the biological knowledge about *Clausena lansium* EOs, which could be exploited particularly for the treatment and prevention of fungal infections caused by *Candida* spp.

## 4. Materials and Methods

### 4.1. Samples and Chemicals

The leaves and fruits of *Clausena lansium* were collected from Haikou City in Hainan, China, in June 2018. Figure 3 shows the leaves and fruits of HLW (a) and CHW (b). All samples were identified by Professor Jianping Tian and were deposited in Public Research Center, Hainan Medical University (Hainan, China). Table 1 lists the sources of *C. lansium* employed in this study. C8–C40 *n*-alkanes (500 mg·L^−1^ in hexane) were purchased from America o2si (Charleston, SC, USA). Other chemicals and their suppliers are listed as follows: agar powder (BeijingSolarbio Sci. & Tech. Co., Ltd., Beijing, China), glucose (Tianjin Fu Chen Chemical Reagent Factory, Tianjin, China), tryptone (Guangdong Huankai Microbial Sci. & Tech. Co., Ltd., Guangdong, China), MH agar powder (Guangdong Huankai Microbial Sci. & Tech. Co., Ltd., Guangdong, China), fluconazole (FLZ) and amphotericin B (AMB) (Aladdin, Shanghai, China), *n*-hexane (chromatographic purity, Aladdin).

### 4.2. Instrumentation and Methods

#### 4.2.1. Instrumentation

Measurements were carried out on a GC-MS-QP2010-Plus system (Shimadzu, Kyoto, Japan) in electron impact ionization (EI) mode. Data were processed with the Shimadzu GC-MS Solution software. The column was ZB-5MS fused silica (30 m × 2.5 mm; 0.25 μm film thickness) from Phenomenex, Torrance, CA, USA. Other instrumentation included a FW-80 high speed grinder (Tianjin Taisite Instrument Co., Ltd, Tianjin, China), an AL104 electronic balance (METTLER TOLEDO Instrument Co., Ltd., Shanghai, China), Automatic digital gel image analysis system Tanon-4100 (Shanghai Tianlong Technology Co., Ltd., Shanghai, China), SPX-250B-Z biochemical incubator (Shanghai Bo Xun Industrial Co., Ltd., Shanghai, China).

#### 4.2.2. Hydrodistillation

The collected leaves of *Clausena lansium* were dried in shade. After removing pulps and seeds of fruits, pericarps were washed and then dried in an oven at 40–50 °C. The leaves and pericarps were milled into powder with a grinder. The EOs were extracted by hydrodistillation for 2.5–3.0 h in a Clevenger-type apparatus. The obtained EOs were stored at 4 °C in an air-tight container and dried using anhydrous sodium sulfate before being analyzed.

#### 4.2.3. GC-MS Analysis

Samples were analyzed by the GC-MS EI method with column ZB-5MS. Helium was used as a carrier gas at a flow rate of 1.0 mL·min^−1^ in split mode (1:20). The column temperature was maintained at 60 °C for 5 min and then programmed to 120 °C at a heating rate of 10 °C·min^−1^, further increased to 170 °C at a rate of 2 °C·min^−1^, and finally increased to 210 °C at a rate of 10 °C·min^−1^ for remaining 10 min. Temperatures of both injector and connector were maintained at 270 °C. Operating parameters of MS were: EI mode at 70 eV with a mass scanning range of 50–500 amu and source temperature of 250 °C. C8–C40 *n*-alkanes were used as reference points in the calculation of relative retention indexes (RIs). The percentages of compositions were obtained from the electronic integration of peak areas. The identity of each compound was determined by comparing its RI to *n*-alkanes and the NIST library database (NIST08 and NIST08s). Percentage of composition was computed from peak areas without applying correction factors.

#### 4.2.4. Candida Strains and Culture Media

The *Candida* strains include the standard reference strains (*C. albicans* ATCC 10231, *C. parapsilosis* ATCC 22019, *C. krusei* ATCC 6258, *C. tropicalis* CMCC(F) c2f, and *C. glabrata* CMCC(F) c6e), and the clinical strains (*C. albicans* 53, *C. albicans* 27). The clinical strains were from oral mucosa of patients with oral candidiasis in the Hainan Wenchang General Hospital located in Wenchang, China. Two clinical isolates were previously identified based on ITS gene sequence analysis. The obtained yeast isolates were stored at −80 °C freezer until use. All strains were streaked onto plates containing the sabouraud dextrose agar (SDA) plate at 35 °C for 24 h. SDA plate composed of 2% (*w*/*v*) agar powder, 2% glucose, and 1% tryptone. A single colony was streaked again to ensure the viability and purity of the strains and the colonies were incubated at 35 °C for 24 h. The colonies with the diameter of about 1 mm were selected and microbial suspensions were prepared in a saline solution. The MH agar plate was used for testing antifungal susceptibilities by the filter paper disc diffusion method.

#### 4.2.5. Probing the Antifungal Activity of the EO Sample by the Filter Paper Disc Diffusion Method

The antifungal activity of the EOs of *C. lansium* was first investigated using the filter paper disc diffusion method following the protocol described in CLSI M44-A2 [25]. Microbial suspensions were prepared in a saline solution and standardized to a turbidity equivalent to that of the tube No. 0.5 on the McFarland scale, corresponding to approximately 1–5 × 10^6^ CFU·mL^−1^. The antifungal drug FLZ (2.5 mg·mL^−1^) and AMB (1.0 mg·mL^−1^), which are commonly used to treat candidiasis, were used as positive controls. 10 μL of *Clausena lansium* EOs and positive controls was respectively added onto a piece of paper placed on the medium in a plate containing the fungal lawn. The plates were incubated at 35 °C for 24 to 48 h, respectively. The antifungal activity the samples was evaluated by measuring zones of inhibition of fungal growth surrounding the paper discs. The zones of inhibition were measured with antibiotic zone scale (Cecon-Brasil) in mm. All the experiments were carried out in triplicate. When the inhibition zone diameters of FLZ and AMB against *Ca* ATCC 10231 were within the prescribed range 28–39 mm and 20–27 mm, respectively, it is considered that the operation of this test was effective.

## Figures and Tables

**Figure 1 molecules-24-01394-f001:**
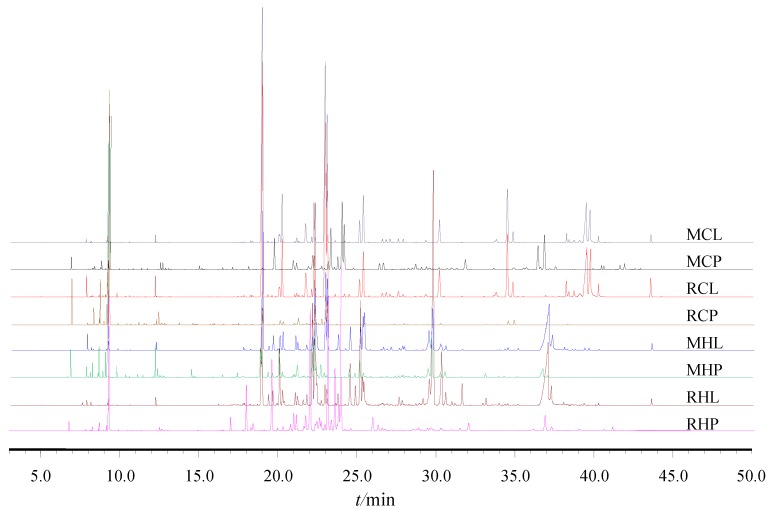
GC-MS chromatograms of *Clausena lansium* EOs.

**Figure 2 molecules-24-01394-f002:**
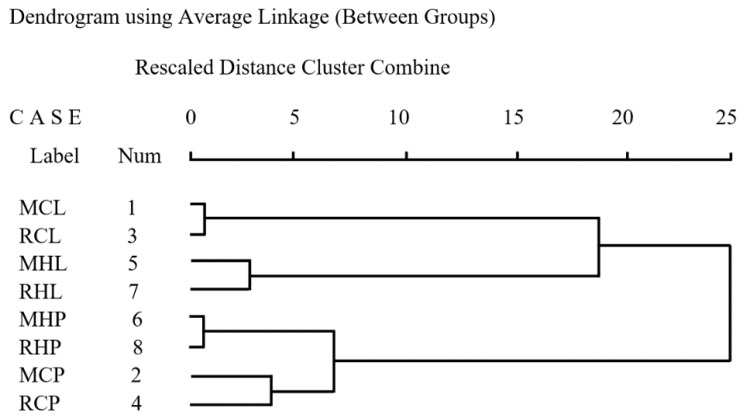
Dendrograms based on the chemical components and contents of *Clausena lansium.* Num: number.

**Figure 3 molecules-24-01394-f003:**
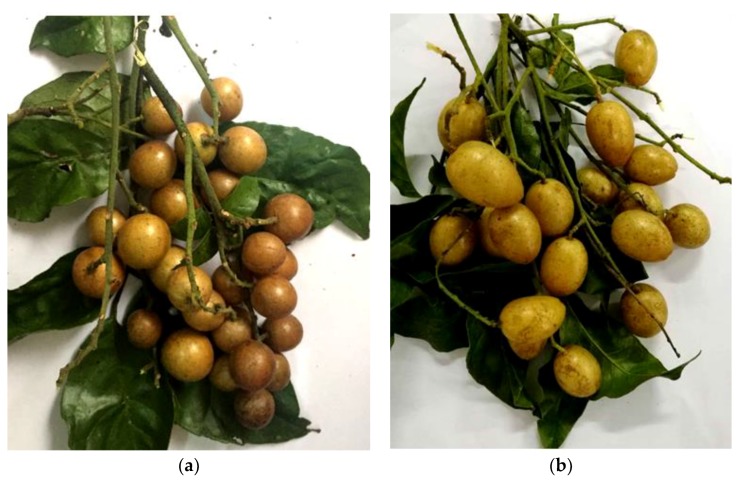
Leaves and ripe fruits of Hainan local wampee (HLW) (**a**) and chicken heart wampee (CHW) (**b**).

**Table 1 molecules-24-01394-t001:** Sources of *Clausena lansium* samples and extraction results of EOs.

No.	Source	Cultivar	Part	Color	Yield ^a^ (%, *v*/*w*)
MCL	Meixiao Village, Haikou City	CHW	Leaf	yellow	0.23 ± 0.08
MCP	Meixiao Village, Haikou City	CHW	Pericarp	light yellow	0.48 ± 0.10
RCL	Rulin Village, Haikou City	CHW	Leaf	yellow	0.37 ± 0.04
RCP	Rulin Village, Haikou City	CHW	Pericarp	light yellow	0.37 ± 0.09
MHL	Meixiao Village, Haikou City	HLW	Leaf	yellow	0.47 ± 0.08
MHP	Meixiao Village, Haikou City	HLW	Pericarp	light yellow	0.50 ± 0.07
RHL	Rulin Village, Haikou City	HLW	Leaf	yellow	0.51 ± 0.04
RHP	Rulin Village, Haikou City	HLW	Pericarp	light yellow	0.47 ± 0.05

^a^ Values represent the means of three independent replicates ±SD.

**Table 2 molecules-24-01394-t002:** Chemical components and contents (%) of *Clausena lansium* EOs.

No.	Components	Molecular Formula	RI ^a^	MCL	MCP	RCL	RCP	MHL	MHP	RHL	RHP
1	α-thujene	C10H16		0.01	-	-	0.02	-	-	-	-
2	α-pinene	C10H16		0.01	1.11	-	5.63		2.03	-	0.76
3	benzaldehyde	C7H6O		-	-	-	-	-	-	0.07	-
4	sabinene	C10H16		0.67	-	0.15	0.21	0.73	0.71	0.12	0.13
5	β-pinene	C10H16		-	-	-	0.07	-	-	-	-
6	6-methyl-5-hepten-2-one	C8H14O		0.02	0.11	0.02	0.04	0.12	0.34	0.07	-
7	myrcene	C10H16		0.06	0.26	-	1.83	0.04	0.98	-	-
8	pseudolimonene	C10H16		-	0.13	-	0.69	-	0.31	-	0.10
9	α-phellandrene	C10H16		0.06	0.68	0.01	4.77	0.04	2.05	-	0.75
10	α-terpinene	C10H16		0.04	0.06	-	0.39	0.02	0.42	-	0.04
11	p-cymene	C10H14		0.06	0.85	0.01	2.29	0.05	1.75	-	0.60
12	limonene	C10H16		0.41	-	0.08	-	0.38	-	0.04	-
13	β-phellandrene	C10H16		0.95	32.43	0.24	45.15	1.08	26.13	0.14	23.15
14	benzeneacetaldehyde	C8H8O		0.01	-	0.01	0.03	-	0.68	-	-
15	γ-terpinene	C10H16	808	0.12	0.06	0.01	0.20	0.06	-	-	0.08
16	terpinolene	C10H16	869	0.03	-	-	0.13	0.01	0.19	-	0.03
17	linalool	C10H18O	897	0.05	0.09	0.03	0.10	0.03	0.05	0.01	0.06
18	*cis*-4-thujanol	C10H18O	925	-	0.12	-	0.25	-	0.17	-	-
19	4-terpineol	C10H18O	981	0.68	0.63	0.31	0.67	0.37	2.41	0.20	-
20	melilotal	C9H10O	985	0.04	0.09	0.06	0.07	0.01	-	-	0.33
21	cryptone	C9H14O	988	-	0.65	-	1.61	-	0.70	-	-
22	α-terpineol	C10H18O	996	0.02	0.13	0.01	0.25	-	0.18	-	0.18
23	(−)-*trans*-pinocarveol	C10H16O	1003	-	0.11	0.01	0.20	-	0.11	-	-
24	geraniol	C10H18O	1048	-	-	-	0.26	-	-	-	-
25	phellandral	C10H16O	1078	-	0.38	-	0.25	-	0.73	-	0.06
26	bornyl acetate	C12H20O2	1083	-	0.15	-	0.18	-	0.12	-	-
27	myrtenyl acetate	C12H18O2	1126	-	0.21	-	0.18	-	-	-	0.03
28	α-cubebene	C15H24	1145	-	0.27	-	0.16	-	0.26	0.01	0.88
29	geranyl ethanoate	C12H20O2	1173	-	-	-	0.07	-	0.07	-	0.09
30	α-copaene	C15H24	1177	0.06	0.39	0.02	0.27	-	0.47	0.04	0.62
31	*trans-*β-bergamotene	C15H24	1186	0.03	-	0.04	-	0.22	-	0.08	0.37
32	β-elemene	C15H24	1189	0.09	-	0.06	0.07	0.03	-	0.08	0.40
33	zingiberene	C15H24	1202	0.10	0.25	0.13	0.15	0.03	-	0.04	-
34	α-cedrene	C15H24	1206	0.10	-	0.12	-	-	-	-	0.05
35	(−)-*trans*-α-bergamotene	C15H24	1212	-	-	-	0.03	0.01	-	0.03	-
36	elixene	C15H24	1218	-	-	-	-	1.66	3.92	2.50	-
37	β-caryophyllene	C15H24	1220	21.13	4.99	23.48	3.36	12.72	2.98	5.46	2.80
38	(+)-α-funebrene	C15H24	1230	0.13	-	0.13	0.03	0.33	0.70	0.46	0.26
39	7,11-dimethyl-3-methylene-1,6,10-dodecatriene	C15H24	1237	0.09	-	0.08	-	1.15	0.59	0.67	-
40	nerylacetone	C13H22O	1243	0.05	-	0.01	0.25	0.02	0.19	0.01	0.16
41	β-santalene	C15H24	1244	-	-	-	-	0.14	-	0.11	-
42	(*E*)-β-farnesene	C15H24	1249	1.12	1.86	1.15	1.00	1.37	3.60	2.62	4.04
43	α-humulene	C15H24	1253	3.75	-	3.74	-	1.64	0.79	0.78	0.74
44	α-himachalene	C15H24	1256	-	1.26	-	0.75	0.08	0.26	0.19	0.60
45	(−)-β-cadinene	C15H24	1270	-	-	-	-	-	0.08	-	-
46	γ-muurolene	C15H24	1275	0.13	0.57	0.11	0.37	-	0.34	0.01	0.32
47	β-himachalene	C15H24	1278	-	0.16	-	0.10	1.27	0.42	0.60	1.12
48	curcumene	C15H24	1280	0.28	2.08	0.37	1.79	0.73	2.21	0.57	3.14
49	(*Z*)-β-farnesene	C15H24	1281	0.13	-	0.13	-	0.11	-	0.02	-
50	β-selinene	C15H24	1287	0.13	0.19	0.07	0.11	0.06	0.13	0.26	0.21
51	γ-cadinene	C15H24	1293	2.25	0.72	1.84	0.27	0.57	0.14	-	0.76
52	γ-elemene	C15H24	1294	-	0.29	-	0.14	-	0.14	0.57	0.67
53	α-muurolene	C15H24	1297	0.02	-	-	-	0.02	-	-	0.84
54	α-farnesene	C15H24	1303	0.54	1.44	0.48	1.07	1.25	4.92	5.60	5.41
55	β-bisabolene	C15H24	1306	2.99	5.61	3.46	3.88	5.30	7.74	9.08	10.21
56	(+)-α-longipinene	C15H24	1308	-	-	-	-	3.75	1.62	5.27	-
57	α-amorphene	C15H24	1312	-	0.41	0.01	0.28	0.04	0.28	0.01	0.85
58	δ-cadinene	C15H24	1315	0.04	2.21	0.03	1.33	0.05	1.87	0.23	4.26
59	β-sesquiphellandrene	C15H24	1324	18.52	10.89	21.80	7.35	3.11	0.72	1.05	2.59
60	α-bergamotene	C15H24	1325	9.49	7.09	10.63	5.41	1.88	0.23	0.83	1.02
61	cubinene	C15H24	1330	-	-	-	0.03	-	-	-	0.39
62	α-bisabolene	C15H24	1336	0.16	0.31	0.17	0.14	0.13	-	0.09	0.27
63	acoradien	C15H24	1340	-	-	-	0.02	1.55	0.09	0.58	0.68
64	alloaromadendrene oxide-(1)	C15H24O	1351	0.22	-	0.16	-	0.14	0.08	0.04	-
65	*trans*-nerolidol	C15H26O	1357	0.17	-	0.07	-	2.39	1.08	2.33	0.05
66	dendrolasin	C15H22O	1368	-	-	-	-	-	0.67	1.05	0.64
67	spathulenol	C15H24O	1372	1.33	0.10	1.96	0.17	2.34	2.65	6.38	3.61
68	caryophyllene oxide	C15H24O	1377	3.64	-	4.50	-	7.82	0.85	3.18	0.43
69	Aromadendrene oxide-(1)	C15H24O	1380	-	-	-	0.42	0.11	0.21	0.06	0.09
70	humuleneepoxide II	C15H24O	1405	0.30	0.12	0.34	-	0.33	0.09	0.20	0.10
71	(−)-spathulenol	C15H24O	1413	-	0.32	-	-	-	0.15	0.08	0.03
72	*cis*-Z-α-bisabolene epoxide	C15H24O	1416	0.25	0.47	0.38	0.08	0.38	0.16	0.05	0.20
73	α-eudesmol	C15H26O	1422	0.08	0.31	0.10	0.31	0.02	0.25	-	0.70
74	cubenol	C15H26O	1428	0.45	0.47	0.37	-	0.22	0.28	0.46	0.08
75	(*E*)-nuciferol	C15H22O	1435	0.21	0.57	0.29	0.36	0.36	0.32	0.06	0.47
76	(+)-γ-gurjunene	C15H24	1442	-	0.15	-	-	0.02	0.18	0.05	0.95
77	α-cadinol	C15H26O	1450	0.05	0.51	0.04	0.06	0.05	0.29	0.10	0.64
78	aromadendrene oxide-(2)	C15H24O	1459	0.36	0.16	0.33	-	-	0.14	0.11	0.04
79	β-bisabolene	C15H24	1470	-	0.51	-	-	3.62	2.22	2.45	1.19
80	*cis*-α-santalol	C15H24O	1477	0.08	0.30	0.03	-	14.72	7.67	21.31	6.59
81	α-bisabolol	C15H26O	1486	2.67	0.75	2.68	0.42	1.28	1.09	4.19	1.60
82	α-santalol	C15H24O	1493	0.06	0.16	0.04	-	0.60	1.07	0.93	2.01
83	(*Z*,*E*)-α-farnesene	C15H24	1542	0.07	0.67	0.04	-	-	0.64	0.53	0.11
84	tricyclopentadeca-3,7-dien [8.4.0.1(11,14)]	C15H22O	1557	0.13	0.47	0.11	-	-	-	0.02	0.10
85	ledol	C15H26O	1562	0.56	0.67	0.46	-	0.05	0.16	0.02	-
86	β-santalol	C15H24O	1569	-	-	-	-	0.12	0.14	0.07	-
87	α-sinensal	C15H22O	1573	5.83	-	5.88	-	0.29	-	0.10	-
88	1,3,6,10-farnesatetraen-12-al	C15H22O	1582	1.20	-	1.04	-	0.04	0.13	0.01	-
89	santalol	C15H24O	1641	-	2.45	-	1.11	17.31	2.92	12.24	4.40
90	*trans*-*Z*-α-bisabolene epoxide	C15H24O	1644	-	5.78	-	1.29	2.70	0.34	1.68	1.50
91	phytone	C18H36O	1650	-	-	0.02	-	0.02	-	-	-
92	*trans*-β-santalol	C15H24O	1670	0.03	-	0.01	-	-	-	0.14	-
93	cedr-8-en-13-ol	C15H24O	1669	0.91	0.54	0.66	-	0.05	-	0.04	0.12
94	lanceol	C15H24O	1676	0.39	0.46	0.20	0.06	0.07	-	0.03	0.68
95	linolenyl alcohol	C18H32O	1690	0.51	0.10	0.26	-	0.06	-	-	-
96	farnesyl acetone	C18H30O	1706	0.57	0.69	0.31	-	-	-	0.07	-
97	methyl palmitate	C17H34O2	1725	7.07	0.93	5.36	-	0.18	-	-	-
98	isophytol	C20H40O	1745	5.20	-	3.55	-	-	-	-	-
99	*n*-hexadecanoic acid	C16H32O2	1763	0.73	-	0.39	-	0.07	-	0.08	-
100	methyl elaidate	C19H36O2	1770	0.04	-	-	-	-	-	-	-
101	phytol	C20H40O	1847	1.19	-	0.60	-	0.58	-	0.31	-
Total peak number				83	76	80	72	84	81	88	78
Total identified peak number				69	63	65	63	70	71	69	66
Total identified peak area percentage				98.87	96.90	99.19	98.18	98.10	98.60	96.87	96.39

^a^ RI: calculated Retention Index -: Not detected.

**Table 3 molecules-24-01394-t003:** Variation of some important volatile components (%) of *Clausena lansium* in four clusters.

Components	Cluster I		Cluster II		Cluster III		Cluster IV	
	MCL	RCL	MHL	RHL	MHP	RHP	MCP	RCP
β-caryophyllene	**21.13**	**23.48**	12.72	5.46	2.98	2.80	4.99	3.36
β-sesquiphellandrene	**18.52**	**21.80**	3.11	1.05	0.72	2.59	**10.89**	**7.35**
*cis*-α-santalol	0.08	0.03	**14.72**	**21.31**	7.67	6.59	0.30	0.00
β-santalol	0.00	0.00	**17.31**	**12.24**	2.92	4.40	2.45	1.11
β-phellandrene	0.95	0.24	1.08	0.14	**26.13**	**23.15**	**32.43**	**45.15**
β-bisabolene	2.99	3.46	5.30	9.08	**7.74**	**10.21**	5.61	3.88
α-bergamotene	9.49	10.63	1.88	0.83	0.23	1.02	7.09	5.41

**Table 4 molecules-24-01394-t004:** Antifungal activity of *Clausena lansium* EOs (x ± s).

Candida Strains	Zone of Inhibition (mm) ^a^									
	MCL	RCL	MHL	RHL	MHP	RHP	MCP	RCP	FLZ	AMB
*C. albicans* ATCC 10231	8.0 ± 0.2	9.0 ± 0.2	8.0 ± 0.2	8.8 ± 0.3	12.8 ± 0.3	12.5 ± 0.2	13.5 ± 0.3	13.2 ± 0.3	33.2 ± 0.4	23.6 ± 0.1
*C. tropicalis* CMCC(F) c2f	7.6 ± 0.2	6.5 ± 0.2	7.5 ± 0.1	6.5 ± 0.1	15.0 ± 0.1	15.5 ± 0.3	12.2 ± 0.3	12.4 ± 0.2	35.8 ± 0.4	23.6 ± 0.3
*C. krusei* ATCC 6258	13.0 ± 0.1	13.0 ± 0.2	11.2 ± 0.3	12.0 ± 0.1	16.1 ± 0.3	16.3 ± 0.2	18.3 ± 0.2	19.3 ± 0.2	12.8 ± 0.3	20.5 ± 0.2
*C. parapsilosis* ATCC 22019	11.0 ± 0.2	10.0 ± 0.2	8.4 ± 0.2	7.9 ± 0.3	11.1 ± 0.2	11.4 ± 0.2	12.1 ± 0.2	12.0 ± 0.2	32.4 ± 0.4	23.7 ± 0.2
*C. glabrata* CMCC(F) c6e	22.0 ± 0.3	22.2 ± 0.2	20.0 ± 0.3	20.9 ± 0.2	22.1 ± 0.3	22.0 ± 0.2	22.8 ± 0.3	23.1 ± 0.3	20.2 ± 0.2	22.0 ± 0.2
*C. albicans* 53	8.2 ± 0.2	8.8 ± 0.2	8.1 ± 0.1	8.6 ± 0.3	12.9 ± 0.2	12.4 ± 0.1	14.5 ± 0.3	14.2 ± 0.2	32.2 ± 0.3	23.8 ± 0.2
*C. albicans* 27	13.0 ± 0.2	12.4 ± 0.2	11.5 ± 0.2	11.3 ± 0.2	14.2 ± 0.3	15.3 ± 0.2	14.0 ± 0.2	13.3 ± 0.1	11.0 ± 0.2	8.2 ± 0.2

^a^ Values represent the means of three independent replicates ±SD; EOs concentration 10 μL/disc; ^b^ FLZ: Fluconazole, 25 μg/dis; ^c^ AMB: Amphotericin B, 10 μg/dis.

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
