# Peer review of "Correlation between Chemical Composition and Antifungal Activity of Clausena lansium Essential Oil against Candida spp."

_molecules, 2019, doi:10.3390/molecules24071394_

Round 1

Reviewer 1 Report

An overall English revision by a native-speaker is advised, as numerous mistakes are detectable (i.e. line 13, Abstract: "efficacity" is not a word, the correct form is "efficacy"). Some sentences are missing the verb. I have highlighted some of them in the attached pdf file.

Lines 14-15: "Clausena lansium (Lour.) Skeels is a large and highly diverse genus of plants (...)". Clausena lansium is NOT A GENUS, IT IS A SPECIES. You shoul correct it to "Clausena is a large and highly diverse genus of plants (...)".

Line 42: the low toxicity of the EOs is debatable, as some relatively toxic effects are reported, such as photosensibilization and allergic reactions.

Lines 45-46: "Clausena lansium (...) there are several species". The species is ONE (Clausenia lansium): I guess the authors meant that there are several VARIETIES of it.

Lines 59-60: the authors report the fruits measures but do not provide any indication on which dimension they are talking about (lenght? diameter?)

Lines 101-103: this is perhaps the biggest flaw of the work. You reported that the pericarps were dried at 40-50°C in an oven: this technique is absolutely wrong, as the materials to be used for the essential oil extraction ARE NOT TO BE SUBJECTED TO HOT TREATMENTS PRIOR TO EO EXTRACTION. This is due to two reasons: 1) the thermal treatments induce a volatilization of the EOs prior to the hydrodistillation; 2) the thermal treatments prior the distillation alter the EO composition due to thermal degradation of the compounds. Moreover, the grinding of the samples prior to extraction is not advisable for the same reasons: the samples should be roughly cut, by hands or with scissors/knives, not ground.

Line 150: the extraction yield values are not reliable for what I wrote in the previous point: the thermal treatment performed prior to the hydrodistillation of the samples might have reduced the EO yield due to its volatilization in the oven.

No details were provided by the authors about the software and the algorithm used to perform the statistical analysis.

Table 2: the authors reported the enantiomeric form for compounds 23, 35, 38, 45, 56 and 76. Moreover, compounds 39 and 42 are the same: they are, indeed, synonims, but they were reported as two separate compounds with different l.r.i.

Author Response

Response to Reviewer 1 Comments

Point 1: An overall English revision by a native-speaker is advised, as numerous mistakes are detectable (i.e. line 13, Abstract: "efficacity" is not a word, the correct form is "efficacy"). Some sentences are missing the verb. I have highlighted some of them in the attached pdf file.

Response 1: We thank you for the comment. We have sought a native-speaker from McMaster University who is an expert in this field to help us modify the language. We hope that English language and style of the revised version are satisfactory. Thanks a lot for the highlighted revision in the attached pdf file.

Point 2: Lines 14-15: "Clausena lansium (Lour.) Skeels is a large and highly diverse genus of plants (...)". Clausena lansium is NOT A GENUS, IT IS A SPECIES. You shoul correct it to "Clausena is a large and highly diverse genus of plants (...)".

Response 2: Thank you for the comment. We have corrected it.

Point 3: Line 42: the low toxicity of the EOs is debatable, as some relatively toxic effects are reported, such as photosensibilization and allergic reactions.

Response 3: We thank you for the comment. We have deleted “low toxicity” in the revised manuscript.

Point 4: Lines 45-46: "Clausena lansium (...) there are several species". The species is ONE (Clausenia lansium): I guess the authors meant that there are several VARIETIES of it.

Response 4: We thank you for the comment. We have changed “species” to “varieties” in the revised manuscript.

Point 5: Lines 59-60: the authors report the fruits measures but do not provide any indication on which dimension they are talking about (lenght? diameter?)

Response 5: The fruits sizes were measured in diameter. We have supplemented it in the revised manuscript.

Point 6: Lines 101-103: this is perhaps the biggest flaw of the work. You reported that the pericarps were dried at 40-50°C in an oven: this technique is absolutely wrong, as the materials to be used for the essential oil extraction ARE NOT TO BE SUBJECTED TO HOT TREATMENTS PRIOR TO EO EXTRACTION. This is due to two reasons: 1) the thermal treatments induce a volatilization of the EOs prior to the hydrodistillation; 2) the thermal treatments prior the distillation alter the EO composition due to thermal degradation of the compounds. Moreover, the grinding of the samples prior to extraction is not advisable for the same reasons: the samples should be roughly cut, by hands or with scissors/knives, not ground.

Line 150: the extraction yield values are not reliable for what I wrote in the previous point: the thermal treatment performed prior to the hydrodistillation of the samples might have reduced the EO yield due to its volatilization in the oven.

Response 6: Thank you very much for your suggestions and comments. Sorry for that we have not considered the proper pretreatment methods for samples. In the course of experiment, we found the leaves of Clausena lansium were much more easily dried in shade than the pericarps. If they were placed too long, the pericarps were easy to go rotten. So we dried them at 40-50 °C in an oven. Can they be dried in lower temperature, like 30-40 °C? In addition, now we have known that the samples should be roughly cut, by hands or with scissors/knives, not ground. Thanks a lot.

Point 7: No details were provided by the authors about the software and the algorithm used to perform the statistical analysis.

Response 7: We thank you for the comment. We have supplemented the software and the algorithm used to perform the statistical analysis.

Point 8: Table 2: the authors reported the enantiomeric form for compounds 23, 35, 38, 45, 56 and 76. Moreover, compounds 39 and 42 are the same: they are, indeed, synonims, but they were reported as two separate compounds with different l.r.i.

Response 8: Thank you for the comment. All the compounds including 23, 35, 38, 45, 56 and 76 were identified by comparing their RIs to n-alkanes and the NIST library database (NIST08 and NIST08s). Compounds can be separated according to their boiling points and polarities by gas chromatography. There are some differences in boiling point between enantiomers, so they can be separated by non-chiral column. Moreover, The CAS numbers of compound 39 and 42 are 77129-48-7 and 18794-84-8, respectively. They are two compounds.

We thank you again for your help! We hope that the revised version is satisfactory.

Reviewer 2 Report

The manuscript  Correlation between Chemical Composition and Antifungal Activity of Clausena lansium Essential Oil against Candida spp.represent original research work, and its described old traditionally used medicinal plant and its anti-Candida effects. Essential oil composition of pericarps and leaves used in the work are known, but obtained results showed promising anti Candidal activity. The presentation stile is good, and results are represented correctly, compared to literature data, which are cited properly.

Author Response

Response to Reviewer 2 Comments

Point: The manuscript “Correlation between Chemical Composition and Antifungal Activity of Clausena lansium Essential Oil against Candida spp.” represent original research work, and its described old traditionally used medicinal plant and its anti-Candida effects. Essential oil composition of pericarps and leaves used in the work are known, but obtained results showed promising anti Candidal activity. The presentation stile is good, and results are represented correctly, compared to literature data, which are cited properly.

Response: We thank you for the positive comment!

Reviewer 3 Report

Major and general concerns:

Improve the English.

Did the authors used a Quality control for the technique?

Why the authors used diffusion methods? MIC evaluation could be more informative about the concentration to use.

Minor errors and questions:

C. tropicalis

Abstract:

Lines 60-62:

Reference 11 and after references 15-17. References 12-14 are after 17.

Material and methods:

Lines 121-132:

Improve this section.

Results and discussion:

Line 232:

Suggestion: change the title to Antifungal activity

Line 258: Biaton et al.

Line 260: Candida in italic

Author Response

Response to Reviewer 3 Comments

Point 1: Major and general concerns: Improve the English.

Response 1: Thank you for the comment. We have sought a native-speaker from McMaster University who is an expert in this field to help us modify the language. We hope that English language and style of the revised version are satisfactory.

Point 2: Did the authors used a Quality control for the technique?

Response 2: Thanks for pointing this out. We have used a quality control for the technique. In the diffusion method, the antifungal drug fluconazole (FLZ) (2.5 mg·mL-1) and amphotericin B (AMB) (1.0 mg·mL-1), which are commonly used to treat candidiasis, were used as positive controls. When the inhibition zone diameters of FLZ and AMB against Ca ATCC 10231 were within the prescribed range 28-39 mm and 20-27 mm, respectively, it is considered that the operation of this test was effective. We have added the quality control for the technique in the revised manuscript.

Point 3: Why the authors used diffusion methods? MIC evaluation could be more informative about the concentration to use.

Response 3: We thank you for the comment. The filter paper disc diffusion method is described in CLSI M44-A2. When the antifungal activity is uncertain, diffusion method can be first used to preliminary determine the concentration. We hope to find the correlation between chemical composition and antifungal activity through our research. Then find out some compounds which may be positively correlated to antifungal activity. These results provide the foundation for the following study. Next, we will use MIC evaluation method to test the effects of several pure chemicals such as β-phellandrene, β-sesquiphellandrene and β-bisabolene each by itself to see their effects.

Point 4: Minor errors and questions:

C. tropicalis

Abstract:

Lines 60-62:

Reference 11 and after references 15-17. References 12-14 are after 17.

Material and methods:

Lines 121-132:

Improve this section.

Results and discussion:

Line 232:

Suggestion: change the title to Antifungal activity

Line 258: Biaton et al.

Line 260: Candida in italic

Response 4: Thank you for the comment. We have checked these minor errors and questions throughout the manuscript and made necessary revisions.

We thank you again for your help! We hope that the revised version is satisfactory.

Round 2

Reviewer 3 Report

The manuscript was improved.